# Knowledge-enhanced Multimodal ECG Representation Learning with Arbitrary-Lead Inputs

## Abstract

Recent advancements in multimodal representation learning for electrocardiogram (ECG) have moved onto learning representations by aligning ECG signals with their paired free-text reports. However, current methods often result in suboptimal alignment of ECG signals with their corresponding text reports, thereby limiting diagnostic accuracy. This is primarily due to the complexity and unstructured nature of medical language, which makes it challenging to effectively align ECG signals with the corresponding text reports. Additionally, these methods are unable to handle arbitrary combinations of ECG leads as inputs, which poses a challenge since 12-lead ECGs may not always be available in under-resourced clinical environments. In this work, we propose the **K**nowledge-enhanced **M**ultimodal **E**CG **R**epresentation **L**earning (**K-MERL**) framework to address these challenges. K-MERL leverages large language models (LLMs) to extract structured knowledge from free-text reports, enhancing the effectiveness of ECG multimodal learning. Furthermore, we design a lead-aware ECG encoder to capture lead-specific spatial-temporal characteristics of 12-lead ECGs, with dynamic lead masking. This novel encoder allows our framework to handle arbitrary lead inputs, rather than being limited to a fixed set of full 12 leads, which existing methods necessities. We evaluate K-MERL on six external ECG datasets and demonstrate its superior capability. K-MERL not only outperforms all existing methods in zero-shot classification and linear probing tasks using 12 leads, but also achieves state-of-the-art (SOTA) results in partial-lead settings, with an average improvement of **16%** in AUC score on zero-shot classification compared to previous SOTA multimodal methods[1].

## 1 Introduction

Recent advancements in deep learning have enabled automated classification of cardiovascular disease (CVD) using electrocardiograms (ECGs), one of the most crucial diagnostic tools. However, most methods are supervised, requiring large amounts of annotated data, which is costly and demands prohibitively extensive expert effort in annotation (Liu et al., 2023b; Huang & Yen, 2022). To address this challenge, self-supervised multimodal learning has recently emerged as an effective approach for learning representative ECG features from accompanied free-text clinical reports (Li et al., 2023; Liu et al., 2024; 2023g). To this end, MERL (Liu et al., 2024) recently introduced the first comprehensive benchmark using the largest dataset MIMIC-ECG (Gow et al.) for pretraining, and six datasets (Wagner et al., 2020; Liu et al., 2018; Zheng et al., 2022; 2020) for evaluating downstream task performance, including zero-shot classification and linear probing.

Despite outperforming signal-only self-supervised approaches, multi-modal approaches, including MERL (Liu et al., 2024), still have notable drawbacks: They directly align ECG signals with reports, introducing unnecessary noise due to the free-text nature of the reports, and failing to fully exploit the rich cardiac knowledge contained within the text. Additionally, they encode ECG in a lead-agnostic manner, overlooking the unique spatial and temporal characteristics of the individual 12 ECG leads. Moreover, they require all 12 leads to be available as input, limiting their ability to generalize across different lead combinations. This raises important practical concerns since full 12-lead ECG data is

---

[1]All data and code will be released upon acceptance.

not always available in clinical environments due to factors such as patient mobility issues, the need for rapid assessments in emergencies, and limited resource in pre-hospital care environments (Bray et al., 2021; Swor et al., 2006; Quinn et al., 2020; Nonogi et al., 2008; Kotelnik et al., 2021; Zhang & Frick, 2019; Nonogi et al., 2008).

To overcome the challenges listed above, we make the following contributions: **(1)** We propose a framework dubbed **K**nowledge-enhanced **E**CG **M**ultimodal **R**epresentation **L**earning (**K-MERL**), which extracts cardiac-related entities from free-text ECG reports, converting unstructured reports into structured knowledge to enhance self-supervised ECG multimodal learning. To the best of our knowledge, this is the first work to leverage structured cardiac entities extracted from clinical reports to improve ECG multimodal learning. **(2)** To effectively capture and leverage the lead-specific spatial and temporal characteristics of 12-lead ECGs, we explore various tokenization and positional embedding techniques. In particular, we design *lead-specific tokenization* and *lead-specific spatial positional embeddings*, enabling the framework to capture the distinctiveness of each lead. **(3)** To enable our framework to handle arbitrary combinations of input leads, we introduce a *dynamic lead masking* strategy. In addition, we propose an *independent segment masking* strategy to further capture lead-specific temporal patterns. **(4)** Our K-MERL framework demonstrates superior performance in zero-shot classification and linear probing on multiple downstream datasets in various lead combinations, from a single lead to all 12 leads.

## 2 RELATED WORK

### 2.1 ECG REPRESENTATION LEARNING.

Recently, ECG self-supervised learning (eSSL) has shown promise in learning ECG representations from unannotated signals (Lai et al., 2023; Chen et al., 2020; Sangha et al., 2024). Contrastive methods such as CLOCS (Kiyasseh et al., 2021) and ASTCL (Wang et al., 2023) explore temporal and spatial invariance, while generative techniques (Zhang et al., 2022; Sawano et al., 2022; Na et al., 2023; Jin et al., 2024) focus on masked segment reconstruction. However, both approaches often lack clinical domain knowledge and are limited to single-modality settings, restricting the quality of learned representations.

Multimodal learning has shown success in multiple biomedical applications (Liu et al., 2023a;c; Wan et al., 2023; Liu et al., 2023f;e;d; Chen et al., 2023). However, ECG signals pose unique challenges due to their complex spatial-temporal structure, necessitating well-tailored modeling. As a result, few studies have explored multimodal ECG learning. (Lalam et al., 2023; Yu et al., 2024) demonstrated the effectiveness of combining ECG and EHR data using large language models (LLMs) to rewrite textual reports. However, their work is restricted to private datasets, making reproducing and comparisons challenging. Other works such as (Li et al., 2023; Liu et al., 2023g) explored multimodal ECG learning for zero-shot classification. However, their methods were over simplistic: They align signals with text without sufficiently capturing the distinctiveness of individual ECG leads, and rely on naive category names as prompts, which fail to capture relative patterns, leading to suboptimal performance. Their limited evaluations on small datasets also fall short of fully assessing multimodal ECG learning in real-world scenarios. Additionally, works such as (Zhao et al., 2024; Wan et al., 2024) focus on ECG-to-text generation tasks, but their results are not publicly accessible, making reproducing and comparisons difficult.

MERL (Liu et al., 2024) is the first open-source study to demonstrate the potential of ECG multimodal learning in zero-shot classification and linear probing across diverse datasets. Therefore, we mainly compare our work to MERL. However, like other methods, MERL relies on all 12 ECG leads as input and cannot handle arbitrary lead combinations, limiting its applicability in real-world clinical scenarios where all 12 leads may not always be available (Jahrsdoerfer et al., 2005; Madias, 2003; Fontana et al., 2019; Maheshwari et al., 2014)

### 2.2 KNOWLEDGE ENHANCED MEDICAL MULTIMODAL LEARNING.

Leveraging medical knowledge to improve medical multimodal learning has advanced significantly, particularly in the radiograph domain, with methods like MedKLIP, KAD, and MAVL (Zhang et al., 2023; Wu et al., 2023; Phan et al., 2024). These approaches focus on extracting structured knowledge, such as clinical entities from free-text radiology reports, and using this information as an additional

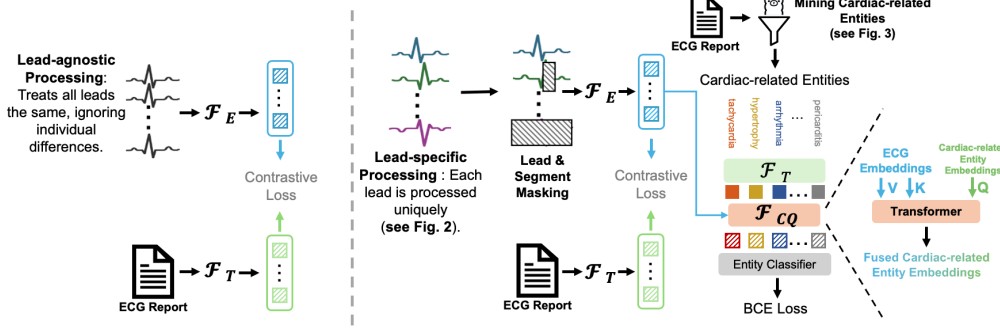

Figure 1: Comparison between classical ECG multimodal learning and our K-MERL framework. **(a):** The classical approaches (e.g., MERL (Liu et al., 2024)) are suboptimal: they processes all leads in a lead-agnostic manner and naively align ECG signals directly free-text reports. **(b):** K-MERL introduces lead-specific processing and lead & segment masking to capture spatial-temporal patterns unique to each lead. It also extracts cardiac-related entities from reports as structured knowledge and aligns them with ECG features to enhance multimodal learning, thereby reducing the complexity introduced by the grammatical structure of free-text reports.

supervisory signal to guide multimodal learning. Many models mimic radiological practices or modify structures based on diagnostic routines (Li et al., 2019; Huang et al., 2020; Zhang et al., 2023; Wu et al., 2023). However, they rely heavily on well-annotated knowledge graphs, such as RadGraph (Delbrouck et al., 2024) and Chest ImaGenome (Wu et al., 2021), which require substantial human annotation and are limited to the radiology domain. Due to the distinct nature of ECG signals compared to radiographs, the above pipelines cannot be directly adapted for ECG multimodal learning. Furthermore, unlike the radiology domain, which has accessible knowledge graphs such as RadGraph, there is no publicly available ECG-specific knowledge graph. As a result, there is no open-source study in ECG yet that utilize medical knowledge to enhance multimodal learning.

### 2.3 CHALLENGE IN PARTIAL LEADS ECG INPUT.

Currently, full 12 leads ECG data dominates publicly accessible ECG datasets (Gow et al.; Ribeiro et al., 2020; Junior et al., 2023). However, in real clinical scenarios, obtaining a standard 12 leads ECG can be excessive and often requires advanced clinical knowledge, which may not always be readily available (Chamadiya et al., 2013; Alizadeh Meghrazi et al., 2020; Dai et al., 2016). This makes partial-lead ECG data both crucial and common for practical applications. Despite its importance, partial leads issue is often overlooked and remain unaddressed in existing ECG multimodal representation learning studies. To handle partial lead inputs across various downstream tasks, in this work, we design lead-specific processing and dynamic lead masking strategies that enable our model to accept any combination of ECG leads as input. adaptable to various clinical scenarios (Jahrsdoerfer et al., 2005; Madias, 2003; Fontana et al., 2019; Maheshwari et al., 2014). We evaluate our model on extensive downstream tasks with partial lead inputs, demonstrating its ability to recognize and adapt to the lead-specific nature of ECG signals.

### 3 METHOD

#### 3.1 OVERVIEW

To this end, we first utilize a general-purpose *open-source* large language model (LLM), such as Llama3.1 (AI@Meta, 2024), without domain-specific fine-tuning, to extract cardiac-related entities from free-text ECG reports.[2] This makes our approach adaptable and well-positioned to benefit from future advancements in LLMs. Additionally, we design a lead-aware ECG encoder with *lead and segment masking* strategies, allowing the model to handle arbitrary lead inputs while capturing lead-specific spatial-temporal patterns.

---

[2]This is based on the well-observed finding that entity extraction is inherently easier than high-level text comprehension in specialized domains and has been shown effective with general-purpose LLMs (Zhang et al., 2023).

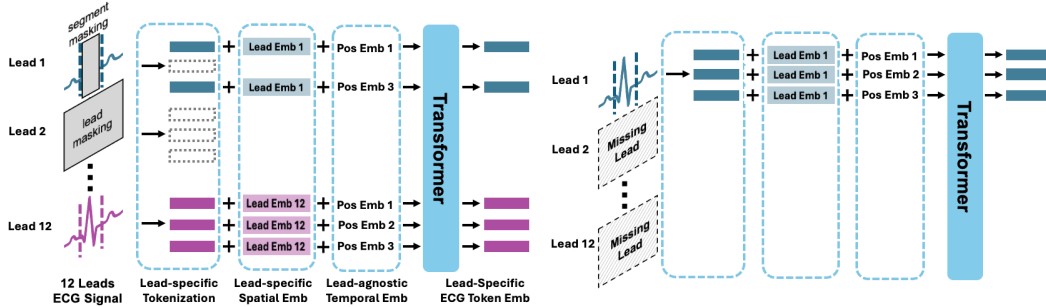

(a) Lead-specific Processing and Masking during Pre-training
(b) Handling Missing Lead Input during Downstream Tasks

Figure 2: Illustration of our lead-specific processing and handling of partial leads input in K-MERL. **(a):** Lead-specific processing and masking during pre-training. The model employs lead-specific tokenization, spatial embeddings, and lead-agnostic temporal embeddings to capture spatial-temporal patterns for each lead (see Sec 3.2). Dynamic lead masking is used to simulate inputs with arbitrary combinations of leads, while segment masking encourage the framework to captures temporal patterns (see Sec 3.3). **(b):** Handling partial lead input during downstream tasks. When leads are missing, the model processes only the available leads using lead-specific embeddings, allowing maintained performance even with incomplete data.

Our overall framework is illustrated in Fig 1(b), shown together with the previous state-of-the-art MERL that is based on naive cross-modal contrastive learning (Liu et al., 2024), in Fig 1(a). While both approaches utilize contrastive learning with an ECG signal encoder $\mathcal{F}_E$ processing signal inputs and a text encoder $\mathcal{F}_T$ processing reports, our method introduces substantial innovations, including lead-specific processing, dynamic masking strategies, and the extraction of cardiac-related entities from free-text reports, significantly enhancing ECG multimodal learning.

In the following sections, we introduce the model framework and lead-specific processing in Sec 3.2, followed by the proposed masking strategies in Sec 3.3. We then describe the pipeline for extracting cardiac-related entities as structured knowledge from ECG reports in Sec 3.4. Finally, in Sec 3.5, we explain the knowledge-enhanced ECG multimodal learning process, a synergy of the aforementioned components.

### 3.2 LEAD-SPECIFIC PROCESSING

To begin with, we define the symbols used in our framework: Given a training dataset $\mathcal{X}$ consisting of $N$ ECG-report pairs, we represent each pair as $(\mathbf{e}_i^l, \mathbf{t}_i)$, where $\mathbf{e}_i^l \in \mathcal{E}$ denotes the raw 12-lead ECG signals for lead $l \in \{1, 2, 3, \dots, 12\}$ of the $i$-th subject ($i = 1, 2, 3, \dots, N$), and $\mathbf{t}_i \in \mathcal{T}$ represents the associated free-text report. We then perform lead-specific processing, as illustrated in Fig 2.

**Lead-specific Tokenization.** Consider an input ECG signal $e_i^l$ with 12 leads and a signal length denoted by $S$. We split the time-series signal into $M$ non-overlapping segments, each segment of length $\frac{S}{M}$, and perform tokenization for them. In this way, each lead ECG is projected into a sequence of tokens:

$$e_i^l[p_1], e_i^l[p_2], e_i^l[p_3], \dots, e_i^l[p_M] \tag{1}$$

where $e_i^l[p_m]$ corresponds to the ECG token for the $m$-th segment for lead $l$. For 12 leads, the total number of tokens is $12 \times M$. Unlike MERL (Liu et al., 2024), which generates a single token for a 12-lead ECG temporal segment, we produce tokens separately for each individual lead to capture the lead-specific nature.

**Lead-specific Spatial Positional Embedding.** We apply a learnable linear projection $\mathbf{W} \in \mathbb{R}^{p \times d}$ to each token $e_i^l[p_m]$. Then, we introduce learnable *lead embeddings* $[\texttt{lead}_1, \dots, \texttt{lead}_{12}]$, where $\texttt{lead}_l \in \mathbb{R}^d$, to capture the characteristics of each lead. The resulting input sequence can be written as:

$$[\texttt{lead}_1 + \mathbf{W}e_i^l[p_1], \dots, \texttt{lead}_1 + \mathbf{W}e_i^l[p_M], \dots, \texttt{lead}_{12} + \mathbf{W}e_i^l[p_1], \dots, \texttt{lead}_{12} + \mathbf{W}e_i^l[p_M]].$$

**Lead-agnostic Temporal Positional Embedding.** In line with lead-specific spatial positional embedding, we also incorporate *learnable lead-agnostic temporal embeddings* to retain the temporal information of ECG signals. These embeddings are denoted as $[\texttt{temp}_1, \dots, \texttt{temp}_M]$, where

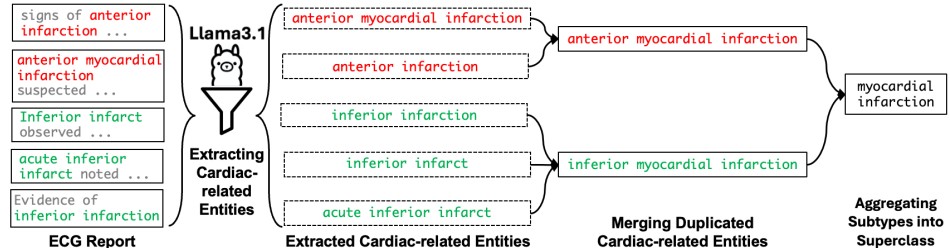

Figure 3: Illustration of mining structured knowledge from free-text reports (see Sec 3.4). First, cardiac-related entities are extracted from free-text ECG reports using an open-source LLM (e.g., Llama3.1-70B-Instruct). Next, we query the LLM to merge duplicated or synonymous cardiac-related entities into a list of unique names. Finally, the LLM detects and aggregates subtypes into their respective superclasses, creating a structured hierarchy of cardiac-related entities.

$\text{temp}_m \in \mathbb{R}^d$. It is worth noting that these positional embeddings are shared across leads, enabling the model to recognize temporal properties across leads, as all leads originate from the same source and share the same temporal domain properties. The resulting input sequence can be written as:

$$[\text{temp}_1 + \text{lead}_1 + We_i^l[p_1], \quad \dots , \quad \text{temp}_M + \text{lead}_1 + We_i^l[p_M],$$
$$\dots , \quad \text{temp}_1 + \text{lead}_{12} + We_i^l[p_1], \quad \dots , \quad \text{temp}_M + \text{lead}_{12} + We_i^l[p_M]].$$

### 3.3 LEAD AND SEGMENT MASKING

Using a fixed number of masked leads limits the model's flexibility in handling arbitrary lead inputs. To address this, we propose **D**ynamic **L**ead **M**asking (**DLM**), enabling the model to handle varying lead combinations (Fig. 2 a). For an ECG signal $e_i^l$ with 12 leads, we first randomly sample a number from $\{9, 10, 11\}$, which determines how many leads will be *masked*. Then, we randomly select a set of unmasked lead indices, denoted as $\hat{l}$, and mask the remaining leads. This approach ensures the model is exposed to diverse combinations of unmasked and masked leads during pretraining. The resulting ECG signal with the selected unmasked leads is denoted as $e_i^{\hat{l}}$.

To better capture the temporal patterns of each ECG lead, we introduce **L**ead-independent **S**egment **M**asking (**LSM**) (Fig. 2 a). Applying masking across all tokens from an ECG signal could lead to imbalances, where some leads have more masked tokens than others. To avoid this, LSM applies masking separately to each lead, ensuring an equal number of masked tokens per lead. For each unmasked lead signal $e_i^{\hat{l}}$, we randomly select masked token indices $\mathcal{H}^{\hat{l}}$ based on a masking proportion of 0.25. The model then processes only the unmasked tokens, denoted as $\{e_i^{\hat{l}}[p_h]\}_{h \notin \mathcal{H}^{\hat{l}}}$.

In the experiments we ablate DLM or LSM to verify their effectiveness, as shown in Tab 2d and Fig 7.

### 3.4 MINING CARDIAC-RELATED ENTITIES FROM REPORT

In this section, we introduce the structured knowledge extraction process for handling free-text ECG reports. The pipeline is illustrated in Fig 3. Since each ECG report provides descriptions of cardiac-related entities, as shown in the leftmost part of Fig. 3, our goal is to extract all positive cardiac-related entities mentioned in the report as structured knowledge to enhance the supervision signals for ECG multimodal learning.

**Extracting Cardiac-related Entities.** Unlike existing biomedical multimodal learning approaches from the radiology domain, which rely on knowledge graphs to extract structured knowledge from reports (Zhang et al., 2023; Wu et al., 2023), we directly query an LLM with the following prompt: 'Please extract all positive Cardiac-related Entities from the given ECG report. Output format is [Entity1, Entity2, ...]'. There are two main reasons for this approach. First, there is no off-the-shelf knowledge graph (KG) specifically focused on ECG, making it impractical to use KG-based methods for extracting structured knowledge. Second, since we are only extracting existing terms from the free-text report, we can easily verify that the extracted cardiac-related entities are present and positive,

ensuring no non-existent terms are generated by the LLM. Moreover, (Zhang et al., 2023) has already demonstrated that a general-purpose LLM can effectively extract existing medical terms from free-text reports independently of any external knowledge database. To ensure accuracy, after each extraction operation, we query the LLM with: 'Please verify the extracted cardiac-related entities as existing and positive in the given report. Output format is YES or NO', and only retain the cardiac-related entities with a 'YES' response. After this stage, we obtain a total of 341 unique cardiac-related entities in the whole dataset..

**Merging Duplicated Cardiac-related Entities.** After extracting all cardiac-related entities from whole dataset, we observe that many names share the same semantics but are expressed differently, as shown in the second part of Fig 3. This variation arises because different clinical protocols generate ECG reports in different styles, even though they describe the same cardiac-related entities. To address this, we query the LLM with: 'Please merge the cardiac-related entities that have the same semantics but different expressions. Here are <all Cardiac-related Entities>. Output format is JSON, where the key is the original name and the value is the merged name.' After this stage, we obtain a total of 252 unique cardiac-related entities in the whole dataset..

**Aggregating Subtypes into Superclasses.** Since cardiac-related entities are organized in a clear hierarchical structure (Arnaout et al., 2016; Okshina et al., 2019), for example, as shown in the rightmost part of Fig 3, 'anterior myocardial infarction' and 'inferior myocardial infarction' are subtypes of the superclass 'Myocardial infarction' (Brieger et al., 2000), we query the LLM with the following prompt: 'Please detect all the superclasses present in <all Cardiac-related Entities>. Output format is JSON, where the key is the superclass name and the values are the cardiac-related entities that belong to this superclass.'

After this stage, we identify 25 superclasses of cardiac-related entities. By the end of the process, we obtain a list of 277 unique cardiac-related entities for the entire dataset. The list of these entities is represented as $\mathcal{Q} = \{q_1, q_2, \ldots, q_Q\}$, where $Q = 277$. For each ECG report $\mathbf{t}_i$, we create a label vector of length 277, where the positions corresponding to present and positive cardiac-related entity are set to 1, and all other positions are set to 0. This results in a binary label vector for each report, which we denote as $\mathbf{y}_i \in \{0, 1\}^{277}$.

### 3.5 KNOWLEDGE-ENHANCED ECG MULTIMODAL LEARNING

**Aligning ECG and Reports.** In this framework, as shown in Fig 1 (b), two distinct encoders for ECG signals and text reports, symbolized as $\mathcal{F}_\text{E}$ and $\mathcal{F}_\text{T}$, transform the sample pair $(\mathbf{e}_i, \mathbf{t}_i)$ into the latent embedding space, represented as $(\mathbf{z}_{e,i}, \mathbf{z}_{t,i})$. The dataset at the feature level is then denoted as $\mathcal{X} = \{(\mathbf{z}_{e,1}, \mathbf{z}_{t,1}), (\mathbf{z}_{e,2}, \mathbf{z}_{t,2}), \ldots, (\mathbf{z}_{e,N}, \mathbf{z}_{t,N})\}$, where $\mathbf{z}_{e,i} = \mathcal{F}_\text{E}(\mathbf{e}_i)$ and $\mathbf{z}_{t,i} = \mathcal{F}_\text{T}(\mathbf{t}_i)$. Afterward, two non-linear projectors for ECG and text embeddings, denoted as $\mathcal{P}_e$ and $\mathcal{P}_t$, transform $\mathbf{z}_{e,i}$ and $\mathbf{z}_{t,i}$ into the same dimensionality $d$, with $\hat{\mathbf{z}}_{e,i} = \mathcal{P}_e(\text{AvgPool}(\mathbf{z}_{e,i}))$ and $\hat{\mathbf{z}}_{t,i} = \mathcal{P}_t(\text{AvgPool}(\mathbf{z}_{t,i}))$. Next, we compute the cosine similarities as $s_{i,i}^{e2t} = \hat{\mathbf{z}}_{e,i}^\top \hat{\mathbf{z}}_{t,i}$, representing the ECG-report similarities, and formulate the ECG-report contrastive loss $\mathcal{L}_\text{contrast}$.

$$\mathcal{L}_{i,j}^{e2t} = -\log \frac{\exp(s_{i,j}^{e2t}/\tau)}{\sum_{k=1}^{L} \mathbb{1}_{[k \neq i]} \exp(s_{i,k}^{e2t}/\eta)}, \quad \mathcal{L}_\text{contrast} = \frac{1}{L} \sum_{i=1}^{N} \sum_{j=1}^{N} \mathcal{L}_{i,j}^{e2t}. \tag{2}$$

The temperature hyper-parameter, denoted as $\eta$, is set to 0.07 in our study. $L$ refers to the batch size per training step, which is a subset of $N$.

**Aligning ECG and Cardiac-related Entities.** To learn the knowledge from extracted cardiac-related entities, we design a cardiac query network, denoted as $\mathcal{F}_\text{CQ}$. This network consists of four transformer layers concatenated with a linear classifier that predicts each ECG's corresponding cardiac entity labels $\mathbf{y}_i$. Given the set of cardiac-related entities $\mathcal{Q}$, we compute a corresponding set of cardiac query vectors using the text encoder, denoted as $\mathbf{Q} = \{\mathbf{q}_1, \mathbf{q}_2, \ldots, \mathbf{q}_Q\}$, where each query vector $\mathbf{q}_i$ is obtained as $\mathbf{q}_i = \mathcal{F}_\text{T}(q_i)$. These query vectors are then used as inputs for the cardiac

query network $\mathcal{F}_{\text{CQ}}$. During pre-training, the ECG features $\mathbf{z}_{e,i}$ serve as the key and value inputs to the cardiac query network $\mathcal{F}_{\text{CQ}}$. We use binary cross-entropy (BCE) loss to compute the predictions from $\mathcal{F}_{\text{CQ}}$ and compare them to the existence labels $\mathbf{y}_i$. The total loss is defined as:

$$\mathcal{L}_{\text{CQ}} = \frac{1}{L} \sum_{i=1}^{N} \text{BCE}(\mathcal{F}_{\text{CQ}}(\mathbf{Q}, \mathbf{z}_{e,i}), \mathbf{y}_i), \quad \mathcal{L}_{\text{total}} = \mathcal{L}_{\text{contrast}} + \mathcal{L}_{\text{CQ}} \tag{3}$$

## 4 EXPERIMENTS

### 4.1 PRE-TRAINING CONFIGURATIONS

**MIMIC-ECG.** We pre-train K-MERL using the MIMIC-ECG dataset (Gow et al.), comprising 800,035 ECG-report pairs. Each sample includes a raw ECG signal recorded at 500Hz over a 10-second duration, along with its corresponding report. For fair comparison with the MERL framework (Liu et al., 2024), we adhere to their preprocessing protocol, available in the official GitHub repository[3]. After preprocessing, we obtain 771,693 samples for model pre-training.

**Implementation.** For pre-training, we inherit the settings from MERL (Liu et al., 2024), using a ViT-tiny model as the ECG encoder and Med-CPT (Jin et al., 2023) as the text encoder. The key differences in our approach are the proposed lead-specific tokenizer and spatial-temporal positional embeddings. For extracting cardiac-related entities from the ECG reports, we utilize Llama3.1-70B-Instruct (AI@Meta, 2024), with ablations of different LLMs shown in Tab 6a. Pre-training configuration details are provided in Sec A.

### 4.2 DOWNSTREAM TASKS CONFIGURATIONS

We evaluate our framework on both zero-shot classification and linear probing, using full and partial lead ECGs across multiple public datasets covering over 100 cardiac conditions. We adhere to the data split and preprocessing provided by MERL (Liu et al., 2024). The tasks are implemented on the following datasets: **(1) PTBXL:** The PTBXL dataset (Wagner et al., 2020) includes 21,837 ECG signals from 18,885 patients, sampled at 500 Hz for 10 seconds. It provides four subsets for multi-label classification: **Superclass** (5 categories), **Subclass** (23 categories), **Form** (19 categories), and **Rhythm** (12 categories), with varying sample sizes. **(2) CPSC2018:** The CPSC2018 dataset (Liu et al., 2018) contains 6,877 12-lead ECG records, sampled at 500 Hz, annotated with 9 distinct labels. **(3) CSN:** The Chapman-Shaoxing-Ningbo (CSN) dataset (Zheng et al., 2020; 2022) comprises 45,152 ECG records sampled at 500 Hz for 10 seconds. After excluding records with 'unknown' annotations, the final curated dataset includes 23,026 ECG records with 38 labels. Detailed information about the datasets used for downstream tasks is presented in Tab 3.

#### 4.2.1 IMPLEMENTATION.

**Zero-shot Classification.** For zero-shot classification, we freeze the entire model and use the original category names from the dataset as entity queries $\mathcal{Q}$ for input to the cardiac query network, $\mathcal{F}_{\text{CQ}}$. The ECG signals are converted into ECG feature with $\mathcal{F}_{\text{E}}$, serving as the key and value inputs for $\mathcal{F}_{\text{CQ}}$. The output of $\mathcal{F}_{\text{CQ}}$ provides the predicted probabilities for each category.

**Linear Probing.** For linear probing, we keep the ECG encoder $\mathcal{F}_{\text{E}}$ frozen and only update the parameters of a randomly initialized linear classifier. We conduct linear probing with $\{1\%, 10\%, 100\%\}$ of the training data. This configuration is used consistently across all linear probing tasks. Further implementation details are provided in the Tab 4.

**Partial Lead Setting.** In the partial lead setting, we follow the lead order from the MIMIC-ECG dataset (Gow et al.): [I, II, III, aVF, aVR, aVL, V1, V2, V3, V4, V5, V6], progressively expanding the input from a single lead to all 12 leads in sequence. In contrast, since MERL (Liu et al., 2024) requires a full 12-lead input, we pad the missing leads with zeros to maintain the 12-lead format.

### 4.3 STATE-OF-THE-ART ON ZERO-SHOT CLASSIFICATION

We first evaluate K-MERL on zero-shot classification using 12-lead input across all downstream datasets. The results for each dataset, along with the average AUC score across six datasets, are

---

[3]https://github.com/cheliu-computation/MERL-ICML2024/tree/main

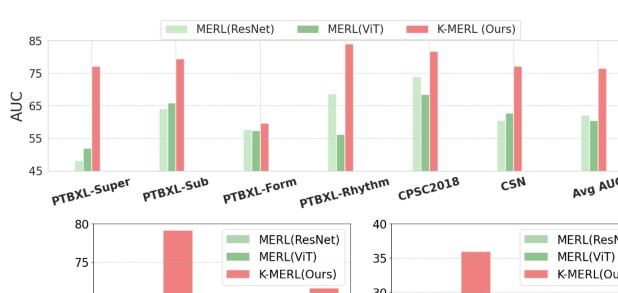

Figure 4: Performance on zero-shot classification across six datasets, comparing K-MERL with previous ECG multimodal learning methods. Notably, we use the original disease category names as prompts for both K-MERL and MERL to ensure a fair comparison.

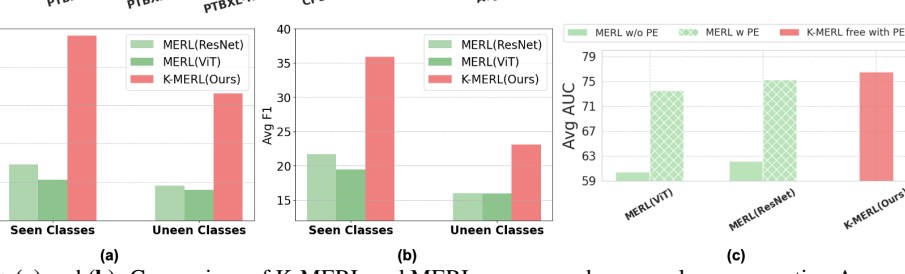

Figure 5: **(a)** and **(b)**: Comparison of K-MERL and MERL on seen and unseen classes, reporting Average AUC and F1 scores. Definitions of seen and unseen classes can be found in Sec 4.3. **(c)**: Comparison of K-MERL and MERL with prompt engineering (PE). Notably, even though MERL with PE uses customized disease prompts with human effort, K-MERL, **free with PE**, still surpasses both versions of MERL, demonstrating its generalizability and effectiveness.

Table 1: Linear probing results of K-MERL and other ECG learning methods, with best results **bolded**.

| Method | PTBXL-Super | | | PTBXL-Sub | | | PTBXL-Form | | | PTBXL-Rhythm | | | CPSC2018 | | | CSN | | |
|---|---|---|---|---|---|---|---|---|---|---|---|---|---|---|---|---|---|---|
| | 1% | 10% | 100% | 1% | 10% | 100% | 1% | 10% | 100% | 1% | 10% | 100% | 1% | 10% | 100% | 1% | 10% | 100% |
| *From Scratch* | | | | | | | | | | | | | | | | | | |
| Random Init (CNN) | 70.45 | 77.09 | 81.61 | 55.82 | 67.60 | 77.91 | 55.82 | 62.54 | 73.00 | 46.26 | 62.36 | 79.29 | 54.96 | 71.47 | 78.33 | 47.22 | 63.17 | 73.13 |
| Random Init (Transformer) | 70.31 | 75.27 | 77.54 | 53.36 | 67.56 | 77.43 | 53.47 | 61.84 | 72.08 | 45.36 | 60.33 | 77.26 | 52.93 | 68.0 | 77.44 | 45.55 | 60.23 | 71.37 |
| *ECG only SSL* | | | | | | | | | | | | | | | | | | |
| SimCLR | 63.41 | 69.77 | 73.53 | 60.84 | 68.27 | 73.39 | 54.98 | 56.97 | 62.52 | 51.41 | 69.44 | 77.73 | 59.78 | 68.52 | 76.54 | 59.02 | 67.26 | 73.20 |
| BYOL | 71.70 | 73.83 | 76.45 | 57.16 | 67.44 | 71.64 | 48.73 | 61.63 | 70.82 | 41.99 | 74.40 | 77.17 | 60.88 | 74.42 | 78.75 | 54.20 | 71.92 | 74.69 |
| BarlowTwins | 72.87 | 75.96 | 78.41 | 62.57 | 70.84 | 74.34 | 52.12 | 60.39 | 66.14 | 50.12 | 73.54 | 77.62 | 55.12 | 72.75 | 78.39 | 60.72 | 71.64 | 77.43 |
| MoCo-v3 | 73.19 | 76.65 | 78.26 | 55.88 | 69.21 | 76.69 | 50.32 | 63.71 | 71.31 | 51.38 | 71.66 | 74.33 | 62.13 | 76.74 | 75.29 | 54.61 | 74.26 | 77.68 |
| SimSiam | 73.15 | 72.70 | 75.63 | 62.52 | 69.31 | 76.38 | 55.16 | 62.91 | 71.31 | 49.30 | 69.47 | 75.92 | 58.35 | 72.89 | 75.31 | 58.25 | 68.61 | 77.41 |
| TS-TCC | 70.73 | 75.88 | 78.91 | 53.54 | 66.98 | 77.87 | 48.04 | 61.79 | 71.18 | 43.34 | 69.48 | 78.23 | 57.07 | 73.62 | 78.72 | 55.26 | 68.48 | 76.79 |
| CLOCS | 68.94 | 73.36 | 76.31 | 57.94 | 72.55 | 76.24 | 51.97 | 57.96 | 72.65 | 47.19 | 71.88 | 76.31 | 59.59 | 77.78 | 77.49 | 54.38 | 71.93 | 76.13 |
| ASTCL | 72.51 | 77.31 | 81.02 | 61.86 | 68.77 | 76.51 | 44.14 | 60.93 | 66.99 | 52.38 | 71.98 | 76.05 | 57.90 | 77.01 | 79.51 | 56.40 | 70.87 | 75.79 |
| CRT | 69.68 | 78.24 | 77.24 | 61.98 | 70.82 | 78.67 | 46.41 | 59.49 | 68.73 | 47.44 | 73.52 | 74.41 | 58.01 | 76.43 | 82.03 | 56.21 | 73.70 | 78.80 |
| ST-MEM | 61.12 | 66.87 | 71.36 | 54.12 | 57.86 | 63.59 | 55.71 | 59.99 | 66.07 | 51.12 | 65.44 | 74.85 | 56.69 | 63.32 | 70.39 | 59.77 | 66.87 | 71.36 |
| *Multimodal Methods* | | | | | | | | | | | | | | | | | | |
| MERL (ResNet) | 82.39 | 86.27 | 88.67 | 64.90 | 80.56 | 84.72 | 58.26 | 72.43 | 79.65 | 53.33 | 82.88 | 88.34 | 70.33 | 85.32 | 90.57 | 66.60 | 82.74 | 87.95 |
| MERL (ViT) | 78.64 | 83.90 | 85.27 | 61.41 | 77.55 | 82.98 | 56.32 | 69.11 | 77.66 | 52.16 | 78.07 | 81.83 | 69.25 | 82.82 | 89.44 | 63.66 | 78.67 | 84.87 |
| **K-MERL (Ours)** | **84.19** | **87.71** | **89.83** | **68.22** | **81.54** | **88.00** | **60.11** | **73.71** | **81.48** | **63.72** | **84.16** | **91.04** | **71.91** | **86.13** | **91.26** | **69.51** | **83.53** | **93.71** |

shown in Fig 4. Our framework significantly outperforms MERL with both backbone architectures, demonstrating the superiority of K-MERL when using the original disease names as text prompts.

It is important to note that MERL heavily relies on prompt engineering (PE), which requires tailoring the text prompt of each possible disease at *inference time*, querying external knowledge bases using LLM, which is inefficient (Liu et al., 2024). To fully showcase the our method's capabilities, we compare K-MERL with the PE-enhanced version of MERL in Fig 5(c). Unlike MERL, K-MERL does not depend on any customized disease prompts at inference time, as it has better leveraged cardiac knowledge contained in the reports during pre-training. Despite being free from PE, K-MERL still surpasses MERL with PE, demonstrating the superiority of our approach.

**State-of-the-art on Unseen Disease Prediction.** Additionally, since we extract cardiac-related entities from reports during pre-training, there may be overlap with categories in downstream tasks. This could provide our model with prior knowledge of certain categories, leading to an unfair comparison with MERL (Liu et al., 2024). To address this, we use Med-CPT (Jin et al., 2023), the text encoder, to extract embeddings for all 277 cardiac-related entities and for all category names in the downstream datasets. We compute the similarity between these embeddings, and if the similarity exceeds 0.95, we consider them overlapped. We identify 35 out of 277 extracted cardiac-related entities that overlap with downstream categories, as listed in Tab 5. We label these as '*Seen Classes*,' while the remaining downstream categories are labeled as '*Unseen Classes*.'

The average AUC for both groups is depicted in Fig 5(a), and the average F1 score in Fig 5(b). K-MERL outperforms MERL in both seen and unseen categories. Notably, both K-MERL and MERL exhibit performance drops on unseen classes compared to seen classes, demonstrating that we successfully detected an overlap of approximately 12.7% between the extracted cardiac-related entities from MIMIC-ECG and downstream categories, effectively separating the tasks into 'seen'

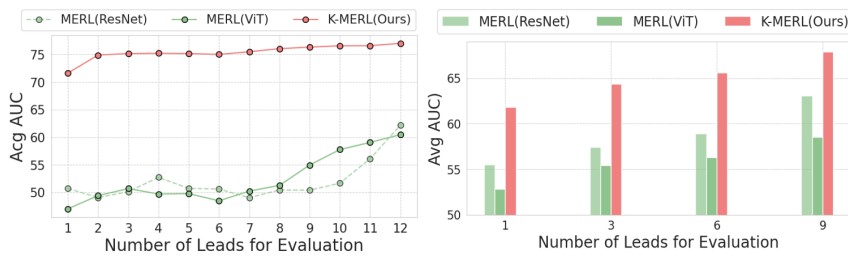

Figure 6: Performance comparison of K-MERL and MERL with partial lead inputs. **(a)** Zero-shot classification shows K-MERL consistently outperforming MERL with two backbones across all lead combinations from 1 to 12. **(b)** Linear probing with 1% data demonstrates K-MERL's superior performance and robustness, even with limited data and varying lead inputs.

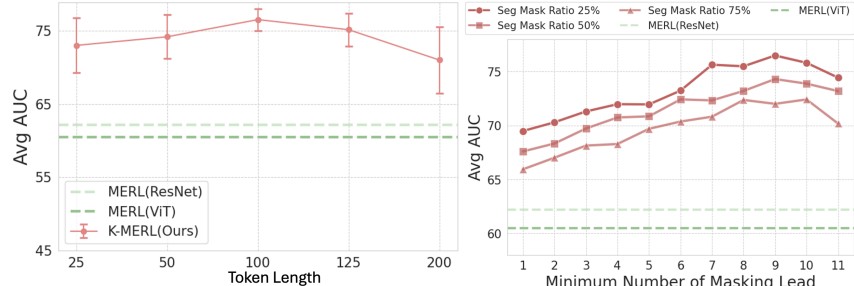

Figure 7: Ablation study on zero-shot classification with 12 leads. **Left:** Performance of K-MERL across varying token lengths, showing optimal results with a token length of 100, consistently outperforming MERL. **Right:** Impact of different segment masking ratios (25%, 50%, 75%) and the minimum number of masked leads. K-MERL outperforms MERL, with the best performance at a 25% mask ratio and a minimum of 9 masked leads.

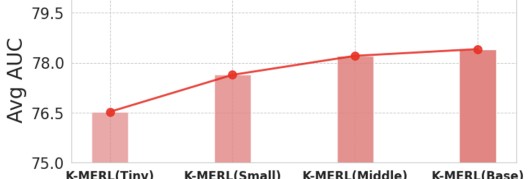

Figure 8: Reported performance of zero-shot classification with scaled ECG encoders. As the model size increases from K-MERL(Tiny) to K-MERL(Base), the performance improves, demonstrating the scalability of the model.

and 'unseen' groups. The results show that K-MERL performs well not only on categories present during pre-training but also on unseen categories, demonstrating its generalizability.

### 4.4 PERFORMANCE OF LINEAR PROBING

As shown in Tab 1, K-MERL consistently outperforms multimodal methods, including MERL (Liu et al., 2024) with both ResNet and ViT backbones, as well as all eSSL methods across datasets and data ratios. This highlights K-MERL's robust performance and the quality of its learned ECG features, which not only improve multimodal tasks but also significantly enhance single-modality tasks.

### 4.5 PERFORMANCE WITH PARTIAL LEADS INPUT

As shown in Fig 6 (a) and (b), K-MERL consistently outperforms MERL across all lead combinations from 1 to 12 in both zero-shot classification and linear probing. Impressively, K-MERL with just a single lead surpasses MERL's performance using all 12 leads. Additionally, K-MERL shows a stable performance trend as the number of leads increases, unlike MERL, which exhibits fluctuations in Fig 6 (a). This demonstrates the effectiveness of our dynamic lead masking strategy, lead-specific processing, and spatial-temporal positional embeddings, contributing to K-MERL's superior results.

## 5 ANALYSIS

This section provides extensive ablation studies on the key components of K-MERL and reports zero-shot classification results for single-lead and 12-lead inputs across all downstream datasets.
**Loss Ablation.** Tab 2a shows the effect of removing $\mathcal{L}_{\text{contrast}}$ and $\mathcal{L}_{\text{CQ}}$ during pre-training. Remov-

Table 2: Results of various ablation experiments. The best results are **bolded**.

(a) Ablating Loss Function.

| Loss | 1 Lead | 12 Leads |
|---|---|---|
| K-MERL (Ours) | **71.61** | **76.52** |
| − ECG-Text Alignment ($\mathcal{L}_{\text{contrast}}$) | 69.23 | 73.98 |
| − ECG-Condition Alignment ($\mathcal{L}_{\text{CQ}}$) | 65.44 | 68.95 |

(b) Effects of Lead-specific Processing.

| Methods | 1 Lead | 12 Leads |
|---|---|---|
| K-MERL (Ours) | **71.61** | **76.52** |
| − Lead-specific Tokenization | 68.47 | 74.23 |
| − Lead-specific Spatial Positional Embedding | 69.12 | 75.35 |
| − Lead-agnostic Temporal Positional Embedding | 70.84 | 75.10 |

(c) Effects of Entities Processing.

| Methods | 1 Lead | 12 Leads |
|---|---|---|
| K-MERL (Ours) | **71.61** | **76.52** |
| − Subtype Aggregation | 70.11 | 74.62 |
| − Merging Duplicated Patterns | 70.54 | 74.93 |

(d) Effects of Masking Strategy.

| Masking Strategy | 1 Lead | 12 Leads |
|---|---|---|
| K-MERL (Ours) | **71.61** | **76.52** |
| − Lead-independent Segment Masking | 70.32 | 75.21 |
| − Segment Masking | 68.93 | 74.74 |
| − Dynamic Lead Masking | 67.84 | 72.11 |
| − Lead Masking | 65.41 | 69.10 |

(e) Effects of Text Encoder.

| Text Encoder | 1 Lead | 12 Leads |
|---|---|---|
| BioClinicalBERT | 68.25 | 73.21 |
| Med-KEBERT | 69.62 | 74.59 |
| Med-CPT | **71.61** | **76.52** |

ing $\mathcal{L}_{\text{CQ}}$, which excludes structured knowledge from cardiac-related entities, leads to a significant performance drop. While removing $\mathcal{L}_{\text{contrast}}$ also reduces performance, the impact is less severe. This indicates that both losses are necessary, with cardiac-related entities alignment providing a larger benefit for pre-training.

**Tokenization Size.** In Fig 7 (a), we ablate the token size $p$ and find the optimal length to be 100. Larger token sizes (e.g., 200) have a more negative impact than smaller sizes (e.g., 25), likely due to convert multiple segments to one token, which introduces ambiguity. Across all token sizes, K-MERL consistently outperforms MERL (Liu et al., 2024), demonstrating the robustness and effectiveness of our method.

**Lead-specific Processing.** In Tab 2b, we ablate the effects of lead-specific tokenization, lead-specific spatial positional embedding, and lead-agnostic temporal embedding. he results show each component enhances K-MERL's performance, with the full combination yielding the best results. The results demonstrate that lead-specific processing is crucial for enabling the ECG multimodal model to recognize lead uniqueness.

**Masking Strategy and Ratio.** Tab 2d shows the results of various masking strategies, where all approaches enhance K-MERL's performance. Removing dynamic lead masking and using a fixed number of masked leads degrades performance, highlighting its importance. Similarly, omitting lead masking during pre-training causes a sharp drop in zero-shot classification, indicating its role in capturing lead-specific features. Fig 7 (b) explores mask ratios and lead masking. An optimal configuration is identified with a mask ratio of 25% and a minimum of 9 masked leads. Increasing the mask ratio beyond this or using more than 9 leads as the minimum for masking leads to a decrease in performance.

**Cardiac-related Entities Processing.** As shown in Tab 2c, both subtype aggregation and merging duplicate entity names improve K-MERL's performance. However, the best results are achieved when both procedures are applied together, indicating they complement each other.

**Text Encoder.** Tab. 2e shows Med-CPT (Jin et al., 2023) outperforms BioClinicalBERT (Alsentzer et al., 2019) and Med-KEBERT (Zhang et al., 2023), due to contrastive pretraining on a large medical corpus, suggesting contrastive pretraining improves text encoder performance for this task.

**Scalability.** We scale our ECG encoder using ViT-Tiny, ViT-Small, ViT-Middle, and ViT-Base, as shown in Fig. 8. K-MERL consistently improves as model size increases, demonstrating its scalability for ECG multimodal learning.

# 6  CONCLUSION

We present K-MERL, a knowledge-enhanced ECG multimodal learning framework capable of processing arbitrary lead inputs. First, we mine cardiac-related entities as structured knowledge from ECG free-text reports using a general LLM, without relying on external domain-specific resources. Next, we align ECG features with these cardiac-related entities to integrate this knowledge into the ECG multimodal learning. Additionally, we introduce lead-specific processing and lead&segment masking strategies to capture the spatial-temporal patterns unique to each ECG lead, enabling the model to handle varying lead inputs. Our experiments on six downstream ECG classification tasks, along with extensive ablation studies, demonstrate K-MERL's superior zero-shot and linear probing performance compared to existing ECG multimodal and self-supervised learning methods.

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

# A   PRE-TRAINING CONFIGURATION

Following MERL (Liu et al., 2024), we employ the AdamW optimizer with a learning rate of $2 \times 10^{-4}$ and a weight decay of $1 \times 10^{-5}$. Pre-training runs for 50 epochs, with a cosine annealing scheduler for learning rate adjustments. We use a batch size of 512 per GPU, with all experiments conducted on eight NVIDIA A100-80GB GPUs.

# B   DOWNSTREAM TASK DETAILS

## B.1   DOWNSTREAM TASK DATA SPLIT

We detail the data splits in Tab. 3. For all datasets, we follow the splits provided by MERL[4]. The preprocessing for all datasets is also done using MERL's official codebase[5].

Table 3: Details on Data Split.

| Dataset | Number of Categories | Train | Valid | Test |
|---|---|---|---|---|
| PTBXL-Super (Wagner et al., 2020) | 5 | 17,084 | 2,146 | 2,158 |
| PTBXL-Sub (Wagner et al., 2020) | 23 | 17,084 | 2,146 | 2,158 |
| PTBXL-Form (Wagner et al., 2020) | 19 | 7,197 | 901 | 880 |
| PTBXL-Rhythm (Wagner et al., 2020) | 12 | 16,832 | 2,100 | 2,098 |
| CPSC2018 (Liu et al., 2018) | 9 | 4,950 | 551 | 1,376 |
| CSN (Zheng et al., 2022; 2020) | 38 | 16,546 | 1,860 | 4,620 |

## B.2   DOWNSTREAM TASK CONFIGURATION

We detail the key hyperparameters used across all downstream tasks in Tab. 4. For each dataset (PTBXL-Super, PTBXL-Sub, PTBXL-Form, PTBXL-Rhythm, CPSC2018, and CSN), we maintain consistency in the learning rate, batch size, number of epochs, and optimizer configuration with MERL (Liu et al., 2024).

Table 4: Hyperparameter settings on downstream tasks.

| | PTBXL-Super | PTBXL-Sub | PTBXL-Form | PTBXL-Rhythm | CPSC2018 | CSN |
|---|---|---|---|---|---|---|
| Learning rate | 0.001 | 0.001 | 0.001 | 0.001 | 0.001 | 0.001 |
| Batch size | 16 | 16 | 16 | 16 | 16 | 16 |
| Epochs | 100 | 100 | 100 | 100 | 100 | 100 |
| Optimizer | AdamW | AdamW | AdamW | AdamW | AdamW | AdamW |
| Learing rate scheduler | Cosine anealing | Cosine anealing | Cosine anealing | Cosine anealing | Cosine anealing | Cosine anealing |
| Warump steps | 5 | 5 | 5 | 5 | 5 | 5 |

## B.3   OVERLAPPED CATEGORIES

As described in Sec 4.3 and Fig 5, we observe that 35 categories are present in both the pre-training and downstream datasets, and we list all the class names in Tab 5.

# C   ADDITIONAL ABLATION STUDIES

Tab 6a, 6b, and 6c present the results of additional ablation studies. (1) Tab 6a shows the impact of various LLMs on processing cardiac-related entities, with Llama3.1-70B-Instruct achieving the best performance across both 1-lead and 12-lead settings. The performance increases with larger LLMs, suggesting that larger models improve cardiac-related entities extraction. (2) Tab 6b explores the effects of different numbers of transformer layers in the Cardiac Query Network $\mathcal{F}_{CQ}$, showing that performance improves as the number of layers increases and saturates at 4 layers. (3) Tab 6c examines the effect of the number of attention heads in $\mathcal{F}_{CQ}$, with 4 heads providing the best performance.

---

[4]https://github.com/cheliu-computation/MERL-ICML2024/tree/main/finetune/data_split

[5]https://github.com/cheliu-computation/MERL-ICML2024/tree/main/finetune

Table 5: Overlap of cardiac-related entities between downstream tasks and the pretraining dataset.

| | |
|---|---|
| prolonged qt interval | normal |
| arrhythmia | first degree av block |
| anterior myocardial infarction | ventricular premature complex |
| conduction disturbance | second degree av block |
| hypertrophy | st depression |
| atrial premature complex | prolonged pr interval |
| t wave abnormalities | premature complex |
| atrial fibrillation | sinus tachycardia |
| sinus arrhythmia | sinus bradycardia |
| atrial flutter | supraventricular tachycardia |
| atrial premature complex | abnormal q wave |
| av block | left bundle branch block |
| myocardial infarction | right bundle branch block |
| st elevation | st-t changes |
| t wave changes | ventricular bigeminy |
| ventricular premature complex | sinus tachycardia |
| atrial flutter | supraventricular tachycardia |
| atrial tachycardia | |

Table 6: Additional Ablation Studies.

(a) Effects of LLM on Processing Cardiac-related Entities.

| Methods | 1 Lead | 12 Leads |
|---|---|---|
| Llama3.1-8B-Instruct | 68.52 | 74.19 |
| Gemma-2-9B | 68.94 | 74.47 |
| Gemma-2-27B | 70.54 | 75.81 |
| Llama3.1-70B-Instruct | **71.61** | **76.52** |

(b) Effects of the Number of Transformer Layers in the Cardiac Query Network $\mathcal{F}_{\text{CQ}}$

| Num of Layers | 1 Lead | 12 Leads |
|---|---|---|
| 1 | 69.92 | 72.96 |
| 2 | 70.14 | 73.13 |
| 3 | 70.31 | 74.40 |
| 4 | **71.61** | **76.52** |
| 5 | 69.25 | 74.94 |

(c) Effects of the Number of Heads in the Cardiac Query Network $\mathcal{F}_{\text{CQ}}$.

| Num of Heads | 1 Lead | 12 Leads |
|---|---|---|
| 1 | 68.76 | 74.89 |
| 2 | 70.25 | 74.23 |
| 3 | 70.27 | 75.36 |
| 4 | **71.61** | **76.52** |
| 5 | 71.23 | 75.48 |