# OpenReview forum: "Knowledge-enhanced Multimodal ECG Representation Learning with Arbitrary-Lead Inputs"
_ICLR.cc/2025/Conference — ICLR 2025 Conference Withdrawn Submission_

### Official Review · Reviewer_NPuw · 2024-10-24

**Soundness:** 3
**Presentation:** 3
**Contribution:** 3
**Rating:** 6
**Confidence:** 4

**Summary:**

This paper proposes a cross-modal foundation model training method that incorporates external knowledge. It generates representations of signal data and text reports through encoders, and then aligns them using contrastive learning. To enhance the model’s robustness, the authors introduce a dynamic signal masking scheme. Subsequently, the model uses LLM to structure the text reports, allowing it to participate in multi-label prediction tasks during pretraining. This strategy significantly expands the supervisory labels for the model, leading to a more effective foundation model that improves performance in downstream applications.

**Strengths:**

The main strength of this study is that it achieves SOTA performance in the ECG classification task, particularly by achieving good performance with relatively small amounts of data. Additionally, it seems that this structured report-based approach using large models is applicable to any medical imaging/signal classification tasks, demonstrating good generalizability. Moreover, the experiments in this paper are very solid.

**Weaknesses:**

This paper has two main shortcomings. The first is that the handling of ECG-entity alignment appears somewhat rough. The masking of the ECG signals is done randomly, which means that it is possible for the part of the signal corresponding to an entity was coincidentally masked. This could negatively affect the performance of knowledge alignment. I believe this is an obvious potential issue, but it seems that the authors have not addressed it in any way. The second issue is that the formatting of the paper is somewhat disorganized, and it appears that in an effort to compress the paper to within 10 pages, the authors removed a lot of content. This makes the technical section seem incomplete, and I had to rely on the figures to guess some implementation details. If this paper is accepted, I hope the authors can restore this content and place some of the experiments in the supplementary materials.

**Questions:**

I am quite curious whether the authors anticipated the issue that the signal segment required for knowledge alignment could potentially be masked, thus introducing noise. How did they handle this issue?

---

### Official Review · Reviewer_t5BL · 2024-10-29

**Soundness:** 3
**Presentation:** 1
**Contribution:** 2
**Rating:** 3
**Confidence:** 4

**Summary:**

The authors present a new method called K-MERL to align ECG recording with ECG reports to enhance evaluation of ECGs. The method is agnostic to the number of leads used and trained in self-supervised way. Comparison with other methods are favorably to the presented method.

**Strengths:**

- Aligning the ECG features with the ECG reports is a meaningful problem and the authors manage to improve this alignment by extracting cardiac related entites from the reports.
- A rigorous comparison a large set of method is performed
- Analysis with ablations to all relevant model components is presented

**Weaknesses:**

- It is unclear why lead masking is clinically useful in the way it is performed in the paper since not all combinations of leads are measured.
- The method is an improvement over MERL but it the results over MERL (especially in Table 1) needs discussion since they seems incremental.
- It is unclear why alignment with the entities helps the model since this information is practically already in the ECG report. This seems to be the main novelty of the method.

**Questions:**

Main questions:
- Line 84: Citing 5 works from the same group of authors Liu et al., 2023a,c,d,e,f for "biomedical applications" is excessive and could be interpreted as inflating citations. Other relevant work could include [1,2,3] and many more. Similarly, line 40, citation of Liu et al., 2023b seems to be unrelated to annotation effort and therefore does not support this claim. Can you justify why this group of authors is most relevant to support these claims in that excess compared to other citations?
- Section 3.3:
  - You mention to mask {9,10,11} leads. Only later it becomes clear that they are only partially masked. Please clarify this.
  - Why do you mask up to 11 leads? In real applications, you will encounter only a small set of variations in ECG lead recordings. This includes 1 lead ECG with specific leads measured and 5 and 6 lead ECG with only specific leads measured. Thus having all random permutations of masked leads is not a realistic setup that you will encounter.
  - Do you use all 12 leads? If so, why? This is redundant since only 8 leads in a 12 lead ECG are mathematically independent.
  - Line 247: In Figure 2 it is unclear whether masking occurs over a complete lead or only on a segment of a lead. Please clarify
  - Line 248: Why is it problematic if "some leads have more tokens than others"? But above you say you mask up to 11 leads meaning that one lead will not be masked at all. This is not clear.
  - Line 251: Why do you choose the masking ratio as 0.25, what is the rationale? Does this mean that 25% of each lead or of all tokens are masked? Please justify and clarify.
- Line 273: Why do you use an LLM to check for the true availability of extracted entities? This can again introduce errors. Instead, you can simply check if the text includes these entities with a string compare operation. What is the performance difference between a string match and your LLM approach or why do you opt for the LLMS approach?
- Eq. 2/3: Can you justify why you need to algin the ECG to both, the report and the entities? Since the entities are originated in the report, there is no additional new information.
- Table 1: The performance gains over MERL (ResNet) are small. This needs discussion on why this is the case and why K-MERL should be preferred. Can you provide evidence why in linear probing your method is not improving significantly and what the difference is to your other experiments where you show larger performance improvements?

Minor points:
- Figure 1: unclear notation especially $\mathcal{F}_{xx}$ is not defined. There are two losses (contrastive and BCE) and in the figure it is unclear what each of them does. Please update the figure to be understand from the information given until its first mentioning.
- Figure 2,4,5,6,7 and Table 1: why is the caption font smaller?
- Line 145: remove full stop before "adaptable"
- Line 154: Starting a section with "To this end" makes it unclear what this refers to.
- Line 197: All your experiment ECGs have a length of 10sec, how large do you choose M then? How do you deal with different sampling frequencies?
- Line 268: remove the abbreviation KG. Only used once.
- Line 278: "from [the] whole dataset".
- Line 297: did you check that all 277 entities are meaningful and medically correct entities?
- Line 307: "Afterward[s]"
- Line 308: The projections $\mathcal{P}_{e}$ and $\mathcal{P}_{t}$ are not in Figure 1. Please correct.
- Line 323: Is this $\mathcal{F}_t$ the same as for the contrastive loss? So two losses are operating on this function and one on all others?
- Line 360: If you don't have 4.2.2, then it doesn't make sense to have 4.2.1
- Figure 4: I first tought the caption refers to the figure below it, which is actualy fig 5c. Please modify to make this clearer.
- Figure 4: Avg AUC as a separate bar is misleading since it looks like its own test set. Please change.
- Line 414: writing incomplete. "to full showcase the our method's"
- Figure 5c: What do you mean with "free with PE". Your method does only barely outperform MERL with PE in this setting which needs discussion.
- Figure 6: Not all lead combinations are actually clinically useful. Thus this comparison is ill-posed.
- Line 533: "lead[ ]&[ ]segment". Best to remove the & at all.

---
References:

[1] Stahlschmidt, S.R., Ulfenborg, B. and Synnergren, J., 2022. Multimodal deep learning for biomedical data fusion: a review. Briefings in Bioinformatics, 23(2), p.bbab569

[2] Acosta, J.N., Falcone, G.J., Rajpurkar, P. and Topol, E.J., 2022. Multimodal biomedical AI. Nature Medicine, 28(9), pp.1773-1784

[3] Li, C., Wong, C., Zhang, S., Usuyama, N., Liu, H., Yang, J., Naumann, T., Poon, H. and Gao, J., 2024. Llava-med: Training a large language-and-vision assistant for biomedicine in one day. Advances in Neural Information Processing Systems, 36.

---

### Official Review · Reviewer_ennm · 2024-11-03

**Soundness:** 3
**Presentation:** 3
**Contribution:** 2
**Rating:** 5
**Confidence:** 5

**Summary:**

The paper addresses limitations in existing multimodal electrocardiogram (ECG) representation learning, particularly issues in aligning ECG signals with free-text reports due to the unstructured nature of medical language. The authors propose a new framework, Knowledge-enhanced Multimodal ECG Representation Learning (K-MERL), which utilizes large language models (LLMs) to extract structured cardiac knowledge from free-text reports to improve the ECG learning process.

Key contributions include:

1. Structured Knowledge Extraction: K-MERL converts free-text ECG reports into structured cardiac entities using LLMs, enhancing the quality of multimodal ECG learning.
2. Lead-aware ECG Encoding: A lead-specific encoder captures spatial-temporal characteristics of individual leads with a dynamic lead masking strategy, enabling flexibility in handling arbitrary lead combinations.

**Strengths:**

The paper is well-structured, offering clear explanations of the model components and experimental setup. Each section logically guides readers through the model's rationale and implementation with clarity. The descriptions of key contributions—such as the lead-aware encoder and entity extraction process—are both detailed and accessible.

The research is bolstered by comprehensive experimental results across six external ECG datasets. The authors conduct thorough evaluations, including zero-shot classification, linear probing, and ablation studies. These assessments effectively demonstrate the robustness of the proposed framework and its individual components.

**Weaknesses:**

The primary difference between K-MERL and the previous MERL (Liu et al., 2024) approach is the extraction of entities from cardiac pre-text; however, this alone does not merit a high score in terms of originality and novelty.

Additionally, the experiments used to validate the hypotheses largely replicate those conducted in the MERL (Liu et al., 2024) paper, lacking significant variation or new insights.
For instance, since text reports are harder to obtain compared to raw ECG data, it would be meaningful to conduct experiments to determine how many ECG-text pairs are necessary for this approach to be effective. Additionally, if utilizing diagnoses based on ICD codes yields better results than extracting cardiac entities from the ECG text reports, it would reduce the significance of using text reports. Instead, it would suggest a novel approach that combines ECG data with diagnostic history as a new modality.

Furthermore, while the authors claim that K-MERL enables learning of multimodal ECG representations, the text reports in the MIMIC-ECG dataset are essentially rule-based text diagnoses initially generated by the ECG equipment provider and subsequently reviewed by medical professionals. Thus, these ECG text reports are more a structured representation of ECG features than an entirely new modality (e.g., blood tests, vital signs, or medical images) distinct from the ECG modality itself.

**Questions:**

It would be beneficial for the authors to clearly address the concerns raised above regarding the limited originality compared to MERL, as well as the limitations of considering ECG and text reports as truly multimodal ECG representations.

---

### Official Review · Reviewer_tKuY · 2024-11-03

**Soundness:** 3
**Presentation:** 2
**Contribution:** 3
**Rating:** 5
**Confidence:** 4

**Summary:**

This paper introduces Knowledge-enhanced Multimodal ECG Representation Learning (K-MERL), a new framework that improves ECG diagnostic accuracy by addressing limitations in current ECG multimodal learning models. K-MERL utilizes large language models to extract structured knowledge from free-text ECG reports, enhancing the alignment of ECG signals with textual information. Furthermore, it introduces a lead-aware ECG encoder with dynamic lead masking, enabling the model to handle arbitrary ECG lead combinations, a critical feature for diverse clinical settings. Extensive evaluations on six external datasets demonstrate K-MERL’s superior performance, achieving state-of-the-art results in both zero-shot classification and linear probing, especially in scenarios with partial lead inputs, underscoring its adaptability and robustness in real-world applications.

**Strengths:**

1. The proposed framework is overall reasonable.
2. The idea of tackling lead missing (or arbituray leads) seems clinically relevant and important.
3. The proposed framework achieves good performance on the benchmark datasets.

**Weaknesses:**

1. The rationale and advantages of the "lead-aware ECG encoder" should be further elaborated. The basic idea is to treat signals from different leads as separate tokens and to apply different position embeddings to these leads. While this approach can help learn and distinguish information from various leads, it may also complicate network training due to the increased introduction of raw but noisy signals as input. Therefore, it is essential to further clarify and validate the rationale and advantages of this approach.

2. One of the main contributions is the claim regarding the ability to handle missing leads in deployment. Although the proposed lead random masking strategy is reasonable, its effectiveness and advantages should be further validated. The baseline compared (MERL with zero mean filling) is suboptimal, and it is expected that this baseline would demonstrate inferior performance. It is challenging to assert that the proposed framework achieves its goals based solely on the limited experimental results presented. A better comparison might be MERL with random masking during training.

3. The proposed framework utilizes a fixed classifier for classification. Therefore, it remains unclear whether the claim of "zero-shot classification" is accurate. Moreover, the comparison with the previous state-of-the-art (SOTA) MERL may also be questionable, as MERL adopts a CLIP-based approach for final classification.

4. The entire framework appears to be an ad-hoc combination of several existing components, making the advancements in this field unclear. The authors should further clarify their technical contributions or insights, in addition to presenting the positive experimental results.

**Questions:**

Besides the questions/comments in the above weakness section.  The authors should improve the presentaiton of this paper. For example, in the motivation of this paper, the authors first mentioned suboptimal alignment and then leading modalities, while for the method part, the authors first introduce the method to tackle the second challenging and then the first challenge. Moreover, for section 3.1, the first and second paragraphs seems redudant.

---

### Official Review · Reviewer_uuRN · 2024-11-04

**Soundness:** 3
**Presentation:** 2
**Contribution:** 1
**Rating:** 3
**Confidence:** 5

**Summary:**

The paper aims to address the limitations of MERL by introducing lead-specific processing and leveraging cardiac-related entities extracted from large language models (LLMs) to improve alignment between ECGs and text reports. It evaluates the approach on various downstream datasets, demonstrating superior performance in linear probing and zero-shot classification compared to other baselines.

**Strengths:**

The strength of this paper lies in its innovative integration of lead-specific processing and cardiac-related entities from LLMs to enhance ECG-text alignment, demonstrating superior performance across various downstream tasks.

**Weaknesses:**

1. **Contribution Clarity and Experimental Validation**:
   - The paper highlights improvements over MERL, specifically in alignment between ECGs and free-text reports and a method considering spatio-temporal aspect of ECGs. However, there are no experiments demonstrating superior alignment in K-MERL compared to MERL, nor is there an ablation study showing that using LAMA for cardiac entity extraction results in less noise.
   - The spatio-temporal aspect is not convincingly validated. While partial lead input experiments are presented, they are insufficient. Comparisons with other spatio-temporal methods are missing.

2. **Presentation Issues**:
   - The token size \( p \) is not mentioned in the tokenization section.
   - Figure 7 lacks clear labeling between parts (a) and (b).
   - Figures are poorly aligned with the text, placed too closely together, and not self-contained, particularly Table 2.
   - The inclusion of baseline performance in Figure 7 for ablation results is unnecessary and could have been presented more effectively in a table.
   - The notation for loss calculation is problematic, as \( L \) is the mini-batch size and \( N \) the total dataset size, which seems inappropriate.
   - If I have understood correctly, the formula in Section 3.2 under **Lead-specific Spatial Positional Embedding**, [$lead_{1}$ + **W**$e_{i}^{l}$[$p_{1}$], ..., $lead_{1}$ + **W**$e_{i}^{l}$[$p_{M}$], ..., $lead_{12}$ + **W**$e_{i}^{l}$[$p_{1}$], ... , $lead_{12}$ + **W**$e_{i}^{l}$[$p_{M}$]] should be updated to: [$lead_{1}$ + **W**$e_{i}^{1}$[$p_{1}$], ..., $lead_{1}$ + **W**$e_{i}^{1}$[$p_{M}$], ..., $lead_{12}$ + **W**$e_{i}^{12}$[$p_{1}$], ... , $lead_{12}$ + **W**$e_{i}^{12}$[$p_{M}$]]


3. **Prompt statement**:
   - The prompt statement selection lacks justification, unlike the detailed description provided in MERL.

4. **Missing Performance Metrics**:
   - The paper does not show results for full fine-tuning or partial fine-tuning.
   - Training and inference times compared to MERL are not reported, which is essential for understanding the practical implications of the proposed method.

**Questions:**

1.  **Ambiguities of text encoder**:
    - It is unclear whether the text encoder is fixed or trained during pre-training. The role of the text encoder needs more clarity.
2.  **Ambiguities of lead-specific tokenization**:
    - The lead-specific tokenization process is not well explained. Questions arise on how experiments like those in Table 2(b) were conducted, especially regarding the application of lead-specific spatial positional embedding without lead-specific tokenization.(The experiment which earns 68.47 for 1 Lead and 74.23 for 12 Leads)
3.  **Ambiguities of seen classes and unseen classes**:
    - The zero-shot evaluation lacks clarity on how the 35 fixed classes were determined across different datasets.
4.  **Differnece between other baseline methods**:
    - How does the "lead-specific processing" proposed in this paper differ from the approach used in ST-MEM, which you cited?

---

### Official Review · Reviewer_jhk9 · 2024-11-04

**Soundness:** 3
**Presentation:** 2
**Contribution:** 2
**Rating:** 3
**Confidence:** 4

**Summary:**

The authors propose a new multimodal method for optimizing an encoder to process ECG signals, enhancing the information in the reports associated with such tracing. They claim the limitations of simply aligning the report representations with the ECG tracing representations and incorporating a new loss function that assesses the ability to infer which relevant clinical entities appear in the original report.

Furthermore, they suggest that when processing multi-lead ECG signals with a transformer architecture, lead embedding should be added to increase the learning potential of the model.

**Strengths:**

- Multimodal Learning is a powerful approach. Studies like this one about how to incorporate the information from reports in order to enhance the learning of the model are needed.

- The core of the paper, incorporating this supervised objective based on the clinical entities appearing in the original report makes sense. In addition, they demonstrate that it improves the overall performance of the method compared to other methods.

**Weaknesses:**

- My biggest concern is that IMO the paper is not properly contextualized according to the existing literature. Methods like MedKLIP, KAD, or MAVL are briefly mentioned in line 106 and no further explanation is provided about what makes K-MERL stand out compared with them. I am finding the proposed framework mirrors KAD's framework without adding any novelty to the Multi-Modal framework.

- Authors claim that they design a novel "lead-specific tokenization". (Line 63) I do not see any differences between the way they embed lead information compared with other studies such as ST-MEM paper (Also used as a benchmark), which also includes this kind of lead embedding in its framework.

- Even the amount of baselines used during the evaluation is significant, most of them (all except one) are just trained on single-modality data. In addition, Most of those are not ECG-specific methods but image-processing ones. I am missing some relevant baselines such as PCLR [1], MAEFE [2], or DEAPS [3]. (See references).

[1] Nathaniel Diamant, Erik Reinertsen, Steven Song, Aaron D. Aguirre, Collin M. Stultz, and Puneet
Batra. Patient contrastive learning: A performant, expressive, and practical approach to electro-
cardiogram modeling. PLOS Computational Biology, 18(2):1–16, 02 2022. doi: 10.1371/journal.
pcbi.1009862. URL https://doi.org/10.1371/journal.pcbi.1009862.

[2] Huaicheng Zhang, Wenhan Liu, Jiguang Shi, Sheng Chang, Hao Wang, Jin He, and Qijun Huang.
Maefe: Masked autoencoders family of electrocardiogram for self-supervised pretraining and
transfer learning. IEEE Transactions on Instrumentation and Measurement, 72:1–15, 2023. doi:
10.1109/TIM.2022.3228267.

[3] Adrian Atienza, Jakob Bardram, and Sadasivan Puthusserypady. Contrastive learning is not optimal
for quasiperiodic time series. In Kate Larson (ed.), Proceedings of the Thirty-Third International
Joint Conference on Artificial Intelligence, IJCAI-24, pp. 3661–3668. International Joint Con-
ferences on Artificial Intelligence Organization, 8 2024. doi: 10.24963/ijcai.2024/405. URL
https://doi.org/10.24963/ijcai.2024/405. Main Track.

**Questions:**

- I find the two loss functions redundant. In fact, according to the Ablation Study, it seems that the contrastive loss is of little value. Section 3.5 details the two cost functions, but IMO it is not clear what kind of information is expected to be captured by the Contrastive Loss that cannot be obtained from the ECB loss. Could you please elaborate on this?

- Related to this first question, ECB Loss and Contrastive Loss have different scales, but in the total loss function, no regularisation is applied on these scales. Have you tried to apply some kind of regularisation on the Contrastive Loss, which has wider ranges (especially at the beginning of the pre-training) to mitigate this difference in scales?

- I see you are training every method for 50 epochs. What is the rationale behind it? I am expecting when removing the Contrastive Loss, the method will converge faster and maybe a better weights configuration can be achieved with less amount of epochs and benefit from early stopping according to loss values from a evaluation dataset split.

---

### Note · Authors · 2025-01-29

I have read and agree with the venue's withdrawal policy on behalf of myself and my co-authors.